# Impact of reduced non-Gaussianity on analysis and forecast accuracy by assimilating every-30-second radar observation with ensemble Kalman filter: idealized experiments of deep convection

Arata Amemiya[1,2,*] and Takemasa Miyoshi[1,2]

[1]RIKEN Center for Interdisciplinary Theoretical and Mathematical Sciences (iTHEMS), Wako, Japan
[2]RIKEN Center for Computational Science (R-CCS), Kobe, Japan
[*]Current affiliation: Japan Weather Association, Tokyo, Japan

**Correspondence:** Arata Amemiya (amemiya.arata@jwa.or.jp)

**Abstract.** Impact of reduced non-Gaussianity on analysis and forecast accuracy by assimilating every-30-second radar observation with ensemble Kalman filter: idealized experiments of deep convection

## 1 Introduction

The use of convection-permitting numerical weather prediction (NWP) models is now becoming a standard practice for short-range precipitation forecast. It is complementary to nowcasting based on simple extrapolation of the precipitation image motion or optical flow. The forecast by NWP models is expected to provide a better representation of precipitation than the forecast by optical flow for a lead time beyond a few hours (Sun et al., 2014; Clark et al., 2016).

Important information on precipitation in convection-permitting NWP models is provided mainly by data assimilation of Doppler weather radars. Currently, most operational centers apply a variational data assimilation method at a 1-hour interval, with radar reflectivity incorporated through latent heat nudging (Gustafsson et al., 2018). Alternatively, the ensemble Kalman filter (EnKF) has also been extensively studied because of its low implementation cost without the need for adjoint models. As EnKF favors sequential data assimilation with a shorter interval than the window length of four-dimensional variational data assimilation (4D-Var) (Fertig et al., 2007), it is common to perform data assimilation cycles with the same frequency as the available observation, typically 5 minutes for most current operational weather radars (Schraff et al., 2016). In addition, another possible option is to use the four-dimensional extension of EnKF, which assimilates observations in a time window of a specified interval. It enables us to use the full information of frequent observation while maintaining the longer data assimilation interval.

The non-Gaussianity of error probability distribution is a major problem for the convective scale EnKF with radar data. Here, the strong non-Gaussianity can be caused by nonlinear observation operators and rapid nonlinear evolution of state variables. The non-Gaussianity makes the analysis based on the EnKF suboptimal (Lei et al., 2010). In this study, we are motivated to investigate the impact of non-Gaussianity on analysis and forecast accuracy with convective-permitting NWP.

Assimilating radar data at an interval shorter than 5 minutes can mitigate the non-Gaussianity problem. Previous studies used the phased array weather radar (PAWR) in Japan with observation interval of 30 seconds. They assimilated PAWR data every 30 seconds using the local ensemble transform Kalman filter (LETKF) to capture the rapid growth of localized intense thunderstorms in metropolitan areas of Japan (Miyoshi et al., 2016b, a; Maejima et al., 2017; Honda et al., 2022; Miyoshi et al., 2023). Other studies also performed high-frequency data assimilation with different phased array radar systems (Kuster et al., 2015; Wu et al., 2018; Huang et al., 2020; Stratman et al., 2020; Huang et al., 2022; Palmer et al., 2022).

Some studies have demonstrated the advantage of frequent data assimilation with an interval of less than 5 minutes. Xue et al. (2006) discussed the possible use of the Collaborative Adaptive Sensing of the Atmosphere (CASA) radar system with a frequent observation mode, showing improved analysis accuracy of dynamical variables by assimilating radar data with 1 or 2 minute intervals. Maejima and Miyoshi (2020) studied the impact of changing the window size of the four-dimensional LETKF in the real case observed by Osaka PAWR and reported the advantage of an assimilation interval of less than 3 minutes. Maejima et al. (2022) performed an observing system simulation experiment (OSSE) for a hypothetical network of PAWR in western Japan and showed that the assimilation of radar observation every 30 seconds is more effective than the case with every 5 minutes in predicting the observed heavy rain distribution. Ruiz et al. (2021) investigated the non-Gaussianity by assimilating the Osaka PAWR data using the LETKF with 1000 members and showed a significant reduction of non-Gaussianity and differences of the analysis mean value of vertical winds by the assimilation interval of 30 seconds compared with 1,2 and 5 minutes. However, the previous studies used real-world data without sufficient verification data for unobserved variables. Additionally, in real-world cases, it is difficult to distinguish the effect of non-Gaussianity from other factors which may degrade the analysis and forecast performances, such as the errors in the model physics and observation operators, limited observation coverage, and multi-scale background error structure.

In this study, we perform a series of idealized OSSEs to investigate the impact of assimilating radar observation at very high frequency on the non-Gaussianity, the analysis accuracy for the variables which are not directly observed, and the accuracy of extended ensemble precipitation forecast. We carefully design the OSSEs so that we exclude other complex real-world factors as possible. The findings of this study would provide insights into future designs on convection-permitting NWP with radar data assimilation, even though every 30 seconds radar data assimilation is a very limited practice at this moment. Also, the investigation on the impact of non-Gaussianity would encourage future studies on non-Gaussian data assimilation methods.

This paper is organized as follows. Section 2 describes idealized OSSEs with the assimilation of radar reflectivity every 30 seconds. Section 3 shows the results of the experiments in terms of analyses and extended ensemble forecasts, followed by an investigation on the impact on 30-second data assimilation on the non-Gaussianity of the ensemble perturbation and its impact on the analysis fields. Section 4 discusses the influence of larger-scale uncertainty based on the results of additional experiments. Section 5 provides a conclusion. re

## 2    Methodology

### 2.1    Overall experimental design

We first describe the overall strategy of the OSSEs in this study. We focus on the impact of assimilating radar observation with a very high frequency on the non-Gaussian characteristics of the background error of state variables and the accuracy of analysis and extended forecast. Therefore, in the first set of experiments, we exclude other factors which we usually have in real-world applications and significantly affect the performance of data assimilation.

First, we perform perfect-model OSSEs, in which the nature run is generated by the same model as the one used in data assimilation experiments. We also use the same observation operator for generating synthetic observations to be assimilated and for the verification purpose in the observation space.

Second, we study the ideal case of a single deep convection triggered by a warm bubble in a conditionally unstable atmosphere. This simplifies the causal relationship between the evolutions of hydrometeors and winds. This causal relationship is often complex with multi-scale interactions in the real world. In addition, we assume only convective-scale uncertainty in the initial ensemble by perturbing only warm bubbles while using the same vertical profiles for the initialization of each member.

Third, we use a large ensemble size for ensemble data assimilation to reduce the effect of sampling error. Also, inflation methods are not applied such as additive or multiplicative covariance inflation, relaxation to prior spread (RTPS), and relaxation to prior perturbation (RTPP), even though these applications might improve the analysis accuracy. This enable us to avoid additional modifications to the form of propability distribution function of the ensemble perturbations and to focus more on the impact of analysis frequency on non-Gaussianity.

In addition to this highly idealized set of OSSEs, we also perform complemental experiments with more realistic settings, which apply perturbations in the thermal and wind background vertical profiles. This is closer to a real-world data assimilation problem where the first guess has uncertainty not only at a convective scale but also at larger scales.

### 2.2    Model and the nature run

In this study, we use the Scalable Computing for Advanced Library and Environment Regional Model (SCALE-RM; Nishizawa et al. (2015)) as a NWP model. The SCALE-RM employs a single-moment 6-category cloud microphysics parameterizaion of Tomita (2008), a Smagorinsky-type subgrid-scale turbulence parameterization of Smagorinsky (1963), and the level 2.5 closure of Mellor-Yamada Nakanishi-Niino type boundary layer parameterization (Nakanishi and Niino, 2004). Radiation and convective cloud parameterizations are not used in the experiments of this study.

We simulate the development of a deep convective cloud with a following idealized setting. The computational domain is 3-dimensional and has a 160 km × 160 km horizontal extent with a 16 km model top. The grid spacing is regular, 1 km horizontally and 200 m vertically. Lateral boundary conditions are periodic in both the X and Y directions. Vertical profiles of horizontal wind, temperature, and specific humidity are prescribed to set the initial condition. To trigger a convective cell, a warm bubble is imposed near the center of the domain, namely, at 75 km from western and southern boundary of the domain and 3 km height from the surface. The warm bubble has a positive temperature anomaly with the maximum intensity of 1 K

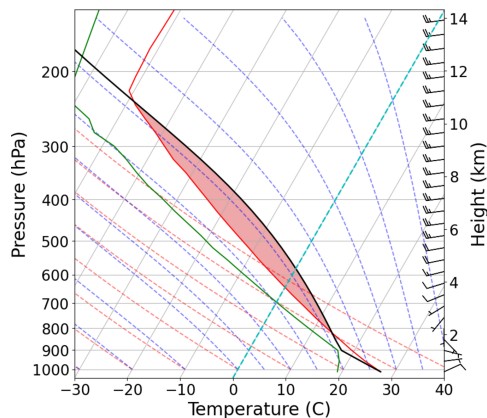

**Figure 1.** A skew-T plot of background atmospheric thermal profile of the experiment. The red shaded area corresponds to the convective available potential energy (CAPE). The profile of horizontal wind is shown on the right side of the figure.

and a 3-dimensional Gaussian shape, with horizontal and vertical length scales of 25 km and 1.5 km, respectively. With these settings, we first performed a single forecast to create a time series of model state variables which is considered the truth. We hereafter call it the "nature run".

Figure 1 shows the skew-T plot of the vertical profile of the initial background state of the atmosphere. It has convection available potential energy (CAPE) of 1655 J kg$^{-1}$ and a strong westerly wind shear. Figure 2 shows the time evolution of the nature run in radar reflectivity calculated by the observation operator, which is described in the next section. The warm bubble triggers an intense deep convective cell that reaches the maximum cloud top height in about an hour. Then the area of strong reflectivity starts to unfold and eventually evolves to two major cells, while the entire system moves eastward.

### 2.3 Data assimilation system and the synthetic observations

We use the data assimilation system known as the SCALE-LETKF, which is the combination of the LETKF with the SCALE-RM. The SCALE-LETKF for radar data assimilation was developed and utilized in previous studies (Miyoshi et al., 2016a, b; Maejima et al., 2017; Amemiya et al., 2020; Honda et al., 2022). The SCALE-LETKF directly assimilates radar reflectivity using the observation operator obtained by the radar simulator using the same particle size distribution settings with the microphysics scheme of the model (Amemiya et al., 2020). The SCALE-LETKF was implemented for successful real-time demonstrations of 30-second refresh NWP in 2020 with 50 ensemble members (Honda et al., 2022) and in 2021 during Tokyo Olympic and Paralympic games with 1000 ensemble members(Miyoshi et al., 2023).

We briefly introduce the calculation procedure of the LETKF, cf. Hunt et al. (2007) for details.

In the LETKF, the Kalman filter calculation is performed in the ensemble subspace formed by the background ensemble perturbations. We denote the background perturbation matrix as $\mathbf{X}^b = [\mathbf{x}^{b(1)} - \overline{\mathbf{x}}^b, \dots, \mathbf{x}^{b(K)} - \overline{\mathbf{x}}^b]$ consists, where $K$ is the

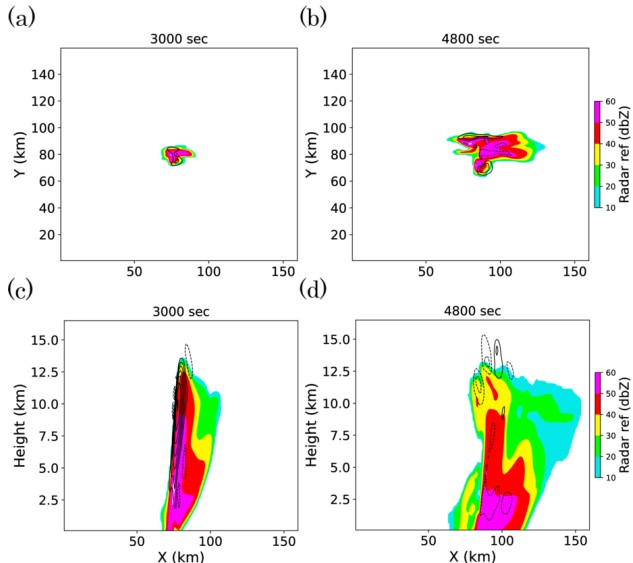

**Figure 2.** Radar reflectivity (color shades) and vertical velocity (thin black contours with an interval of 5 m s$^{-1}$) of the nature run at (a,b) 2 km height level and (c,d) the vertical cross section along Y = 80 km (horizontal black lines in a,b). The snapshots at (a,c) 50 minutes and (b,d) 80 minutes from the initial time are shown.

ensemble size. The background error covariance matrix in the model space is

$$\mathbf{P}^b = \frac{1}{K-1}\mathbf{X}^b(\mathbf{X}^b)^T, \tag{1}$$

and the corresponding expression in the ensemble subspace is

$$\tilde{\mathbf{P}}^b = \frac{1}{K-1}\mathbf{I}. \tag{2}$$

Then the analysis error covariance matrix in the ensemble subspace is

$$\tilde{\mathbf{P}}^a = [(K-1)\mathbf{I} + (\mathbf{Y}^b)^T\mathbf{R}^{-1}\mathbf{Y}^b]^{-1}, \tag{3}$$

where $\mathbf{Y}^b$ is the matrix whose columns are the background ensemble perturbations in the observation space, and $\mathbf{R}$ is the observation error covariance matrix.

For the consistency with Eqs. (1) and (3), the analysis ensemble perturbation is obtained as follows.

$$\mathbf{X}^a = \mathbf{X}^b\mathbf{W}^a \tag{4}$$

$$\mathbf{W}^a = [\tilde{\mathbf{P}}^a]^{1/2}, \tag{5}$$

The analysis ensemble mean is obtained by the Kalman filter formula.

$$\bar{\mathbf{x}}^a = \bar{\mathbf{x}}^b + \mathbf{X}^b\bar{\mathbf{w}}^a, \tag{6}$$

$$\bar{\mathbf{w}}^a = \tilde{\mathbf{P}}^a(\mathbf{Y}^b)^T\mathbf{R}^{-1}(\mathbf{y}^o - \bar{\mathbf{y}}^b), \tag{7}$$

where $\mathbf{y}^o$ is the observation vector and $\mathbf{y}^b$ is the background ensemble mean in the observation space.

Equation (6) is derived from the Kalman filter formula, which calculates the Kalman gain that minimizes the trace of the analysis error covariance matrix. The analysis coincides with the maximum likelihood solution only when the background and observation errors follow Gaussian distributions. Therefore, when the background error is strongly non-Gaussian, the analysis increment would be suboptimal.

Synthetic observation is generated from the time series of the nature run every 30 seconds. In this study, we assume that
the radar reflectivity is observed throughout the domain, at every grid point of 1 km interval both horizontally and vertically. For simplicity, the scanning geometry and the attenuation effect depending on the radar location are not considered. The assimilation of Doppler velocity is also not considered. Radar reflectivity is calculated from the density and hydrometeor mixing ratios of the nature run using the same observation operator used in the SCALE-LETKF, adding random errors from the Gaussian distribution with the standard deviation of 5.0 dBZ.

Following the method of Aksoy et al. (2009), for both observation and each member of the first guess ensemble, reflectivity values below 10 dBZ are adjusted to 5 dBZ and considered as 'no-precipitation' signals, separated from 'precipitation' signals which are equal to or above 10 dBZ. Reflectivity observation is assimilated only where at least one ensemble member of the first guess has 'precipitation' signal at the grid point. The localization with an approximate Gaussian function of Gaspari and Cohn (1999) is applied, with horizontal length scales of 4 km and 2 km for precipitation and no-precipitation observations,
respectively. The vertical length scale is set to 2km for both of them. These settings are the same with Ruiz et al. (2021). We use 100 ensemble members for all experiments, which we consider large enough for this study, with the small localization length scale limiting the effective degrees of freedom of the background error. We set the threshold of the number of members which has a 'precipitation' signal at a grid point for the observed radar reflectivity to be assimilated. We also set the upper limit of the number of observations assimilated at a grid point to 100, which is equal to the ensemble size.

We perform data assimilation experiments with three different configurations, namely, (i) 5-minute 3D-LETKF (hereafter called 5MIN-3D), which uses only the reflectivity observed every 5 minutes, (ii) 5-minute 4D-LETKF (hereafter called 5MIN-4D), which uses observations of every 30 seconds within the 5-minute time window from the previous to the current analysis time, and (iii) 30-second 3D-LETKF (hereafter called 30SEC), which assimilates the observation every 30 seconds. The 5MIN-4D case does not use temporal localization, namely, we set the same weight to all observations in a time window. Therefore, the
5MIN-4D and 30SEC use the same observation information in total, while they differ in the assimilation frequency. Although this choice might not be practical, we prioritize exploring the underlying relationship between assimilation frequency and non-Gaussianity in an idealized setting.

## 2.4 Initial ensemble perturbation

The design of initial ensemble perturbation is crucially important for the data assimilation with the LETKF, as it largely
determines the characteristics of the multi-scale and multi-variable background error covariance. In this study, as mentioned above, we simplify the problem by setting the idealized initial perturbation. We use the same vertical profile with that of the nature run, to initialize the state variables of all the members. We perturb the location and intensity of the warm bubble, with

a spatial scale comparable to the size of the convective cell. Further, we add the small random band-pass-filtered potential temperature perturbation over the entire domain to create the non-zero ensemble spread in the area outside the convective cell.

With these initial perturbations, the first guess ensemble at the time when high reflectivity is first observed is expected to have strongly non-Gaussian perturbations with a spatial structure at a convective scale. The parameters used for these perturbations are summarized in Table 1.

**Table 1.** The initial perturbation properties.

|  | Variable | Perturbation type | mean and standard deviation |
|---|---|---|---|
| Warm bubble | Potential temperature | Maxmimum intensity | Mean: 1 K<br>Std. dev. : 0.2K |
|  |  | Center location | Mean: 75 km in X and Y<br>Std. dev. : 25 km in X and Y |
| Whole domain | Potential temperature | Additive noise | Mean: 0 K<br>Std. dev. : 0.2 K (domain average)<br>3-D bandpass filter: 4 km - 40 km |

## 2.5  Verification methods

For each of the three experiments, we perform data assimilation cycles for 50 minutes from the initial time (corresponding to
Figs. 2a and 2c) and evaluate the analysis. We also perform a 30-minute ensemble forecast from the analysis ensemble at that time step. We focus on this particular analysis time because the maximum value of vertical velocity in the updraft reaches its peak value around $40 \, \mathrm{m \, s^{-1}}$ . We assume that the data assimilation runs long enough from the initial time to make the error and ensemble spread values evolve stably, although the rapid evolution of the convection makes it difficult to see the convergence of those values.

We compare the performances of the analyses and extended forecasts among different data assimilation settings. We particularly focus on the non-Gaussianity of first guess ensemble vertical velocity and the accuracy of analysis mean vertical velocity in comparison with the nature run, as it is thought to be one of the most difficult variables to properly estimate from radar observation (Fabry and Meunier, 2020), and has been shown to have a strong sensitivity to the data assimilation interval in the real-world case (Ruiz et al., 2021). For extended forecasts, we mainly evaluate the ensemble mean accumulated surface
precipitation, as it is the most important forecast variable in practice. To quantify the non-Gaussianity of a univariate probability distribution, we use the Kullback–Leibler divergence (KLD) against the Gaussian distribution having the same mean and variance. As we calculate the KLD from the ensemble of finite size, we approximate it by kernel density estimation. Suppose that we have a standardized ensemble of a variable $x_k$ $(k = 1 \ldots K)$. The estimated probability density using Gaussian kernel

is,

$$p(x) = \frac{1}{Kh} \frac{1}{\sqrt{2\pi}} \sum_{k=1}^{K} \exp\left(\frac{-(x - x_k)^2}{2h^2}\right), \tag{8}$$

where $h$ is a kernel bandwidth, which is determined adaptively as follows (Silverman, 1986).

$$h = 1.059 * K^{-0.2} \tag{9}$$

Then the KLD is calculated numerically as follows.

$$\text{KLD}(P||Q) \sim \sum_{j}^{J} p_j \ln \frac{p_j}{q_j}, \tag{10}$$

where $p_j$ and $q_j$ are the values at the center of each bin, for the estimated density $p(x)$ and the normal distribution $q(x)$ respectively.

Mutual information is used to evaluate the functional relationship between the ensembles of two variables. It is defined for two probabilistic variables $x$ and $y$ as follows.

$$\mathbf{I}[x,y] = \text{KLD}(p(x,y)||p(x)p(y)) \tag{11}$$

$$= -\int \int p(x,y) \ln\left(\frac{p(x)p(y)}{p(x,y)}\right) \tag{12}$$

In this study, it is used to evaluate the nonlinear relationship between two variables in the first guess ensemble. For this purpose, this is applied after removing the linear regression of $y$ on $x$ from the original $y$.

## 3   Results

### 3.1   Analysis and first guess fields in reflectivity and vertical velocity

Figure 3 shows the time series of the total number of assimilated reflectivity observation and averaged analysis ensemble spread in reflectivity for every 5 minutes. The assimilation starts at 00:10:00 as the initial convection develops and reflectivity over 10 dBZ emerges in the observation and the first guess. The first The number of assimilated observation peaks around 00:15:00 to 00:20:00, as a large number of 'no-precipitation' observation is assimilated to suppress the emergence of convection at random locations triggered by initial random perturbation. Except for the initial peak, the number of observations gradually increases as the high reflectivity area extends over time as the convective cell develops. The 5MIN-4D and 30SEC assimilate much larger number of observations than the 5MIN-3D case, as expected. The 5MIN-4D has generally larger number than 30SEC because it often has larger number of grid points where some members show artificial 'precipitation' signals to suppress. The time series of averaged ensemble spread indicates that the analysis largely follows the observation, not causing the filter divergence. The average ensemble spread is not significantly different between 5MIN-4D and 30SEC after 00:50:00. Note that these metrics are for reflectivity, which is a variable directly observed. They could be different for other variables as we examine later.

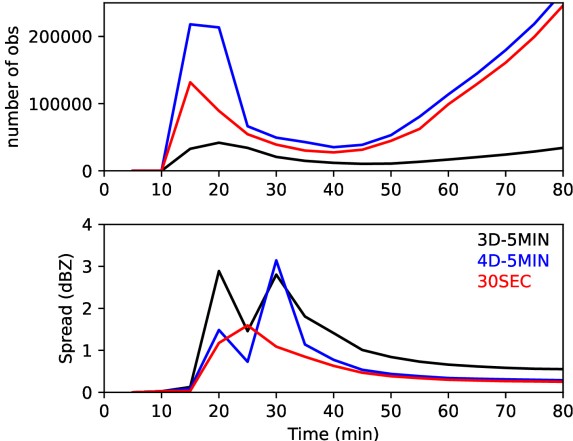

**Figure 3.** Time series of (a) the total number of assimilated observations and (b) the analysis ensemble spread in reflectivity averaged over the grid points where nature run shows values over 10 dBZ. Black, blue, and red curves indicate the results of 5MIN-3D, 5MIN-4D, and 30SEC, respectively.

Figure 4 shows the analysis mean reflectivity at 00:50:00 for 5MIN-3D, 5MIN-4D, and 30SEC. The analysis ensemble mean reflectivity is closely similar to the nature run in all the cases. The ensemble spread of analysis reflectivity has at most 6.6, 5.0, and 3.2 dBZ in the 5MIN-3D, 5MIN-4D, and 30SEC case, respectively. This simply reflects the total number of assimilated observations, with more observations leading to a smaller spread.

Figures 5a-5c compare the analysis mean vertical velocity and its deviation from the nature run. Note that the figures show the area between X = 60 km and 100 km, which is narrower than Fig.4. The 5MIN-3D case underestimates the updraft near its maximum position. The 5MIN-4D has slight errors around the updraft, whereas the 30SEC case almost reproduces the true vertical velocity field. The largest error is located around the maximum of the updraft near 10 km height and not large enough to change the structure of deep convection. The errors in the 5MIN-3D and 5MIN-4D cases are considered to be the remainders of the error in the first guess ensemble shown in Figures 5d and 5e.

Figures 5g-5i show the ensemble spread and the KL divergence in vertical velocity for each case. The case 5MIN-3D shows the largest spread and KL divergence in vertical velocity. The 5MIN-4D shows smaller values, though they are still significantly larger than those of 30SEC. In the case of 5MIN-3D and 5MIN-4D, large values of KL divergence are found near the location of the maximum updraft, the lower troposphere below the updraft, and in the right part of the figures (X = 95 km, Z = 8 km). The area below the updraft is thought to be associated with downdraft caused by precipitation in some members, and the right part may correspond to the border of the convective system. However, there are no significant differences of KL divergence around the area of the maximum updraft between 5MIN-4D and 30SEC cases.

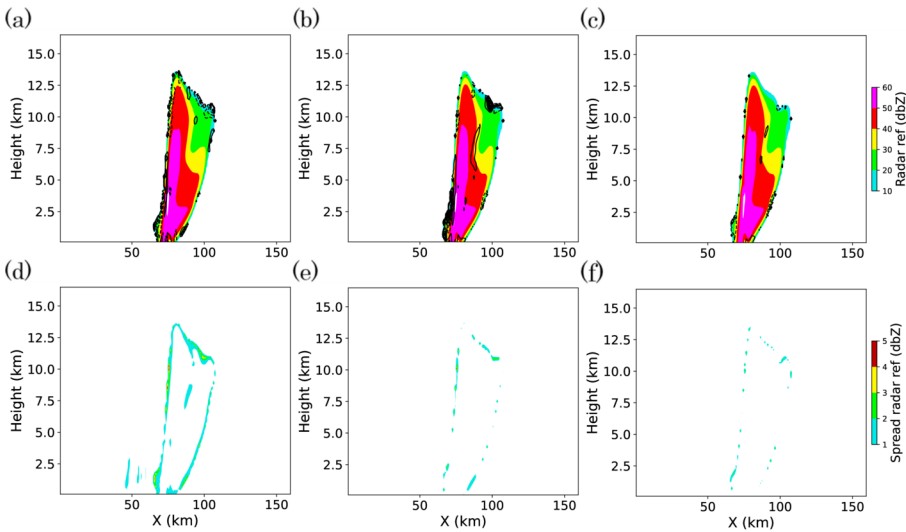

**Figure 4.** Vertical cross sections of (a-c) ensemble mean and (d-f) spread of analysis reflectivity at 00:50:00, for (a,d) 5MIN-3D, (b,e) 5MIN-4D, and (c,f) 30SEC cases respectively. Black contours in (a-c) indicate the difference between analysis mean and nature run, with contour interval of 0.5 dBZ.

## 3.2 Forecast fields in reflectivity and surface precipitation rate

Figures 6a-6c show cross sections of the ensemble mean radar reflectivity of 30-minute forecasts from 00:50:00. The difference
from the nature run is shown in black contours. The three cases show almost similar patterns of large reflectivity area. Some differences among the cases are found in the rear part of the precipitation system, where relatively low reflectivity remains below 4 km and around Y = 70 to 80 km. The 30SEC case shows a higher accuracy in reflectivity in this part, although the difference is small. Figures 6d-6f show the accumulated surface precipitation for 30 minutes between 00:50:00 and 01:20:00 for each case. All of them show a common spatial pattern of error against the nature run. The difference among the cases is
not as significant as the systematic error of the forecasts. In addition, the ensemble spread shown as blue contours indicates the similar spatial pattern and magnitude among the three cases, although the analysis ensemble spreads in reflectivity and vertical velocity shown in Figs. 4 and 5 are significantly different.

## 3.3 Non-Gaussianity and non-linearity

In this section, we examine the error probability distribution and discuss how it impacts on analysis accuracy. Figure 7 shows
the example scatter plots between first guess ensemble of graupel mixing ratio and vertical velocity for each case. The blue markers correspond to the values of each of the 100 ensemble members, indicating the joint probability distribution, and the red markers indicate the nature run. They are extracted from the same grid point (X = 78 km, Y = 80 km, Z = 12.1 km), where the 5MIN-3D case shows large KL divergence of vertical velocity. The 5MIN-3D case shows the joint distribution with a bended shape, whereas the 5MIN-4D and 30SEC cases show the distribution that would be more fitted by a straight line. This is due to

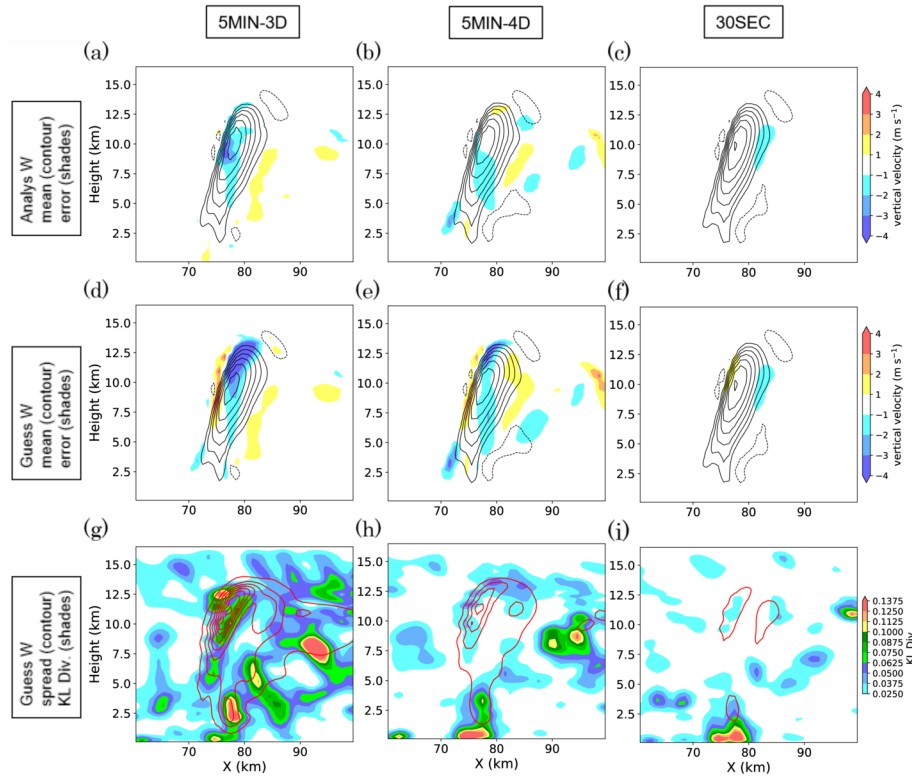

**Figure 5.** Vertical cross sections at Y = 80 km of (a-c) analysis ensemble mean and deviation from the nature run, (d-f) first guess ensemble mean and deviation from the nature run, and (g-i) first guess ensemble spread and KL divergence, of vertical velocity at 00:50:00, for (a,d,g) 5MIN-3D, (b,e,h) 5MIN-4D, and (c,f,i) 30SEC cases respectively. Black contours in (a-f) indicate ensemble mean of vertical velocity with contour interval of 5 m s$^{-1}$. Red contours in (g-i) indicate first guess ensemble spread of vertical velocity with contour interval of 0.5 m s$^{-1}$.

the difference in the ensemble spread as seen in Figures 5d-5f. The rapid error growth produces not only the non-Gaussianity in a single variable but also the nonlinearity in cross-variable relationships in the first guess ensemble. This may be the cause of the analysis error in state variables, as the LETKF calculates the analysis increment using the linear superposition of the ensemble perturbation of the first guess. That treatment is unable to estimate the optimal analysis increment value for both of the two variables when they have a nonlinear functional relationship. To quantify the occurrence of the joint distribution

showing nonlinear relationship such as Figure 7a, we calculate the mutual information between the ensemble members of graupel and vertical velocity at every grid point, after removing the linear dependency. Figure 8 shows the cross section of the mutual information. Large values are found in the upper part of the main convective cell in the 5MIN-3D case, indicating strong nonlinear relationship between the two variables. This tendency of a strongly nonlinear joint distribution between the graupel mixing ratio and the vertical velocity may be one of the factors which contribute to the larger error in analysis mean

vertical velocity in Figure 5a. This approximately nonlinear relationship, as well as large ensemble spread in first guess vertical velocity itself, can introduce suboptimal analysis increment.

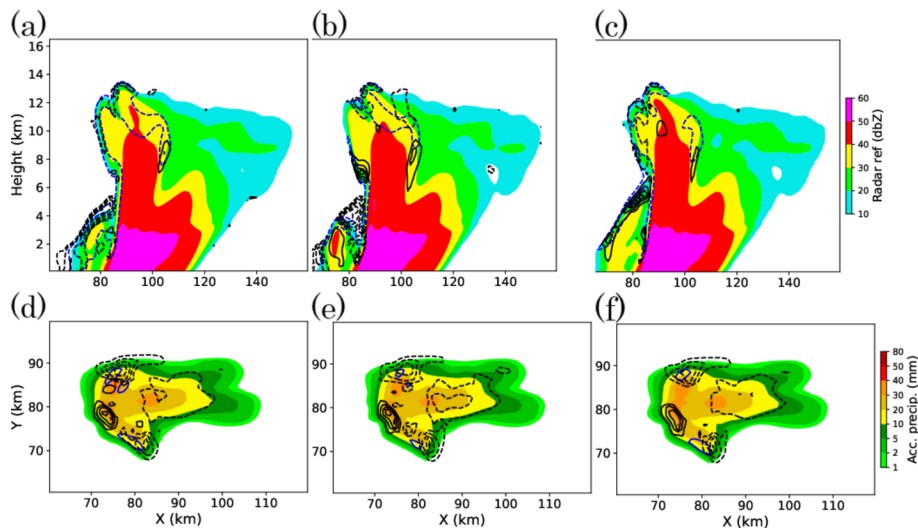

**Figure 6.** (a-c)Vertical cross sections at Y = 80 km of ensemble mean radar reflectivity and (d-f) ensemble mean accumulated surface precipitation of 30 minutes forecast from 00:50:00. They are of (a,d) 5MIN-3D, (b,e) 5MIN-4D, and (c,f) 30SEC cases respectively. Black contours in (a-c) indicate difference from the nature run with a contour interval of 5 dbz. Black contours in (d-f) indicate the difference from the nature run with a contour interval of 2 mm. Blue contours in (a-c) indicate the ensemble spread in reflectivity with a contour interval of 5 dbz. Blue contours in (d-f) indicate the ensemble spread in surface precipitation with a contour interval of 2 mm.

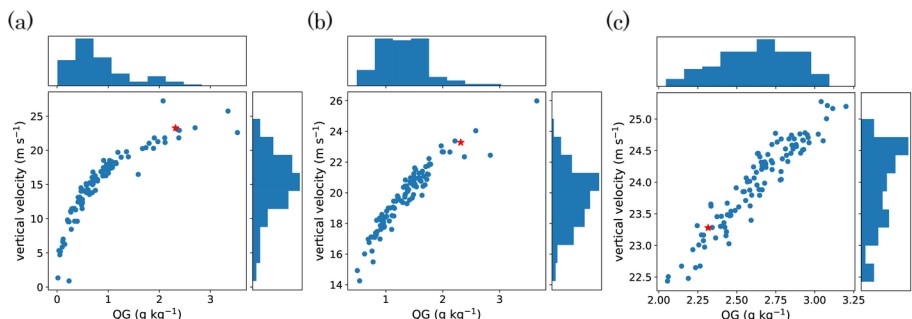

**Figure 7.** The examples of scatter plots of 100 ensemble members between graupel mixing ratio (QG) and vertical velocity (W) for each of (a) 5MIN-3D, (b)5MIN-4D, and (c)30SEC cases, at 00:50:00 as in Fig. 5. The blue dots indicate values of each ensemble member. The red star marker indicates the value of nature run. Histograms of QG and W are shown on the top and on the right of each panel, respectively.

## 4 Additional experiments in the presence of larger-scale errors

### 4.1 Initial ensemble with perturbed background profiles

To extend the discussion in the previous sections to more realistic situations, we conduct additional experiments with back-

ground errors at a larger scale. We perform two additional experiments using the same observation data, one with perturbation

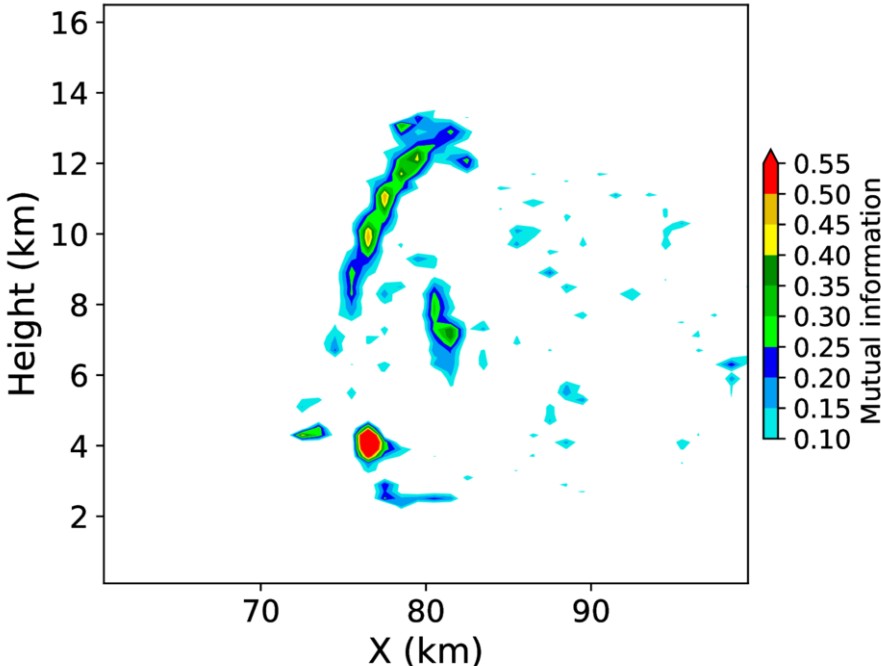

**Figure 8.** Vertical cross section at Y = 80 km of the mutual information between ensemble values of QG and W after removing linear dependency in the 5MIN-3D case.

in the background wind profile, and the other with perturbation in the background thermal profile. In both cases, the background profile of the nature run was perturbed, so that 10 different profiles shown in Fig.9 for each 10 members of the ensemble are used to create a 100-member initial ensemble. The random perturbation described in Section 2.4 (Table 1) is imposed on each ensemble member in the same way as before. Both of those 10 sets include one true profile, indicating only 10 members in the ensemble have accurate background wind or stability profile. The other 90 members are biased and are expected to have significant errors in the evolution at a convective scale.

The same LETKF setting is used in these additional experiments, although the previous setting includes parameters such as small localization scales, which may be suboptimal in the presence of background errors of a larger scale. Our purpose of these experiments is to find implications that help interpret the previous real-world experiment (Ruiz et al., 2021), which was performed with initial and boundary conditions downscaled from the parent model ensemble forecasts with considerable larger-scale uncertainty.

### 4.2 Analysis and first guess in vertical velocity

Figures 10a-10c show the analysis mean, and 10d-10f show the first guess mean vertical velocity in the presence of background wind perturbation. As the ensemble includes members with weaker upper-level wind, they are expected to have convection

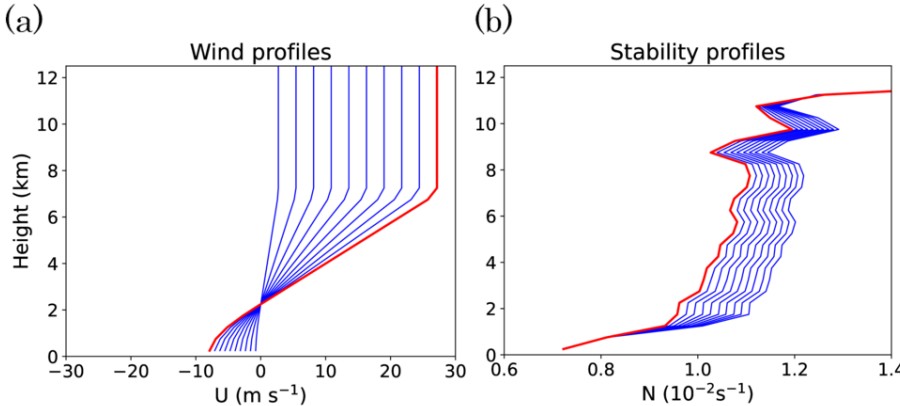

**Figure 9.** Background atmospheric profiles of (a) horizontal wind and (b) atmospheric stability (buoyancy frequency) used in the experiment in Sect. 4. Each panel shows 10 different profiles, which are randomly assigned to the initial condition of 100 members. Red curves indicate the profiles used for the nature run.

location shifted to the left. Consequently, 5MIN-3D and 5MIN-4D cases have a significant dipole pattern of first-guess error in vertical velocity. Additionally, the location shift of the maximum updraft causes a skewed error distribution of vertical velocity in nearby grid points, causing high non-Gaussianity seen in Figures 10g and 10h. The analysis mean of 5MIN-3D has a remaining dipole-shaped error, whereas 5MIN-4D has larger errors with complex spatial pattern. This might be caused by the absence of temporal localization of 4D-LETKF, which assimilates all previous 5-minute observations with equal weight, leading to the suboptimal instantaneous field at the analysis time. The 30SEC case has significantly smaller first guess mean error and KL divergence, leading to smaller analysis mean error.

The results of perturbed stability experiment are more complicated. The first guess mean error in vertical velocity shown in Figures 11d-11f indicates a significant negative bias over the area of updraft in 5MIN-3D and 5MIN-4D cases. This is caused by members with less intense convection under weaker background instability. The KL divergence fields show large values in the upper part of the convection, implying a strongly non-Gaussian distribution among the ensemble. The analysis mean of the 5MIN-3D case has significant remaining negative error in the upper part. Although the background vertical velocity is expected to be correlated with reflectivity, the analysis mean error still remains especially in the upper part, indicating the possibility of a non-Gaussianity effect. On the other hand, the analysis mean vertical velocity of 5MIN-4D has a significantly smaller error than 5MIN-3D in the upper part, whereas it has somewhat larger negative errors in the lower troposphere. The 30SEC case has a smaller first guess error and KL divergence in the area of the updraft. However, the area of larger negative error behind it (around X = 70 km) is found both in the first guess and the analysis mean fields. It might be a side effect of frequent assimilation, possibly triggering artificial convection cells by accumulated unbalanced analysis increments. In summary, in this perturbed stability case, the comparison among different data assimilation frequencies has more complex features, and we have to consider other possible factors than non-Gaussianity, although 5MIN-3D still shows the largest analysis error.

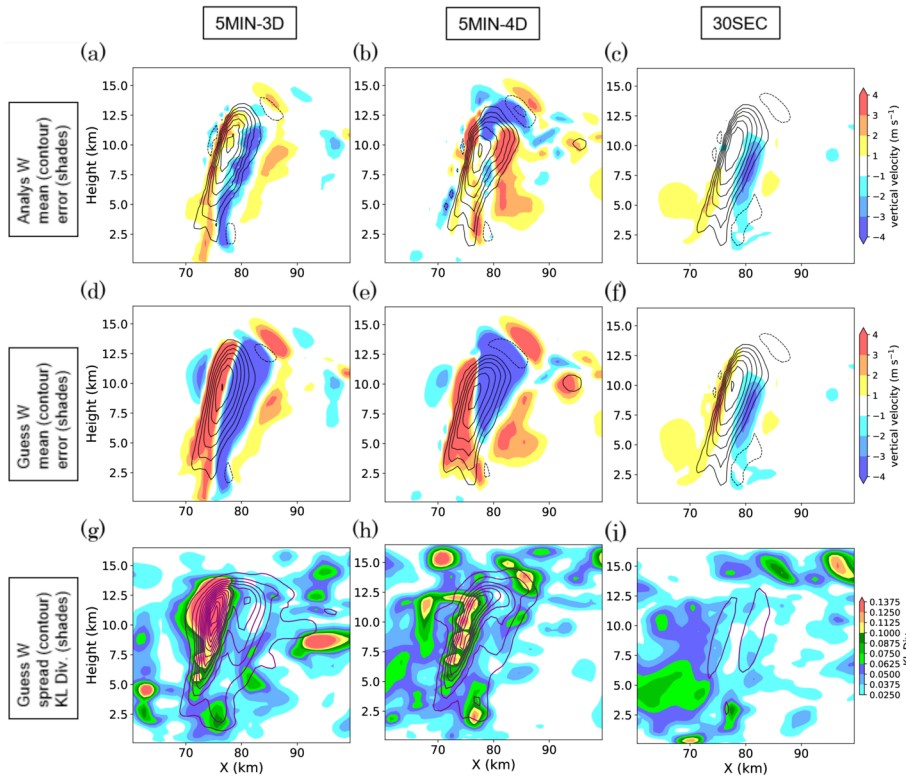

**Figure 10.** Same with figure 4 but for the experiments with background wind perturbation. Purple contours in (g-i) indicate first guess ensemble spread of vertical velocity with contour interval of 1 m s$^{-1}$.

285 **4.3 Forecast in surface precipitation rate**

Finally, we compare the impact of frequent data assimilation on surface precipitation forecast in these experiments with per-turbed background profiles. Figures 12a-12c show the accumulated surface precipitation for 30 minutes between 00:50:00 and 01:20:00 for each of the three cases with perturbed background wind. The common feature of forecast errors among these three cases is that the area of high precipitation is slightly shifted to the left in these figures, as we can see in the areas of large 290 positive errors (X = 70-80 km, Y = 70-80 km). In contrast, the area around the peak of accumulated precipitation (X = 70-90 km, Y = 80-85 km) has negative errors. Compared to these common features, the differences among the three cases have a smaller spatial scale and less significant. Figures 12d-12f show those of the experiments with perturbed background stability. The common error patterns are the higher peak value (X = 80 km, Y = 80 km) and the smaller values in the surrounding area. This is caused by weaker development of convection and concentration of precipitation in a smaller area. These common 295 error patterns in both the perturbed background wind and stability cases coincides with the patterns of the forecast ensemble spread shown in blue contours respectively. In these experiments, the difference among cases with different data assimilation frequency is less significant than the common errors. In overall, in these experiments with perturbed background profiles, the

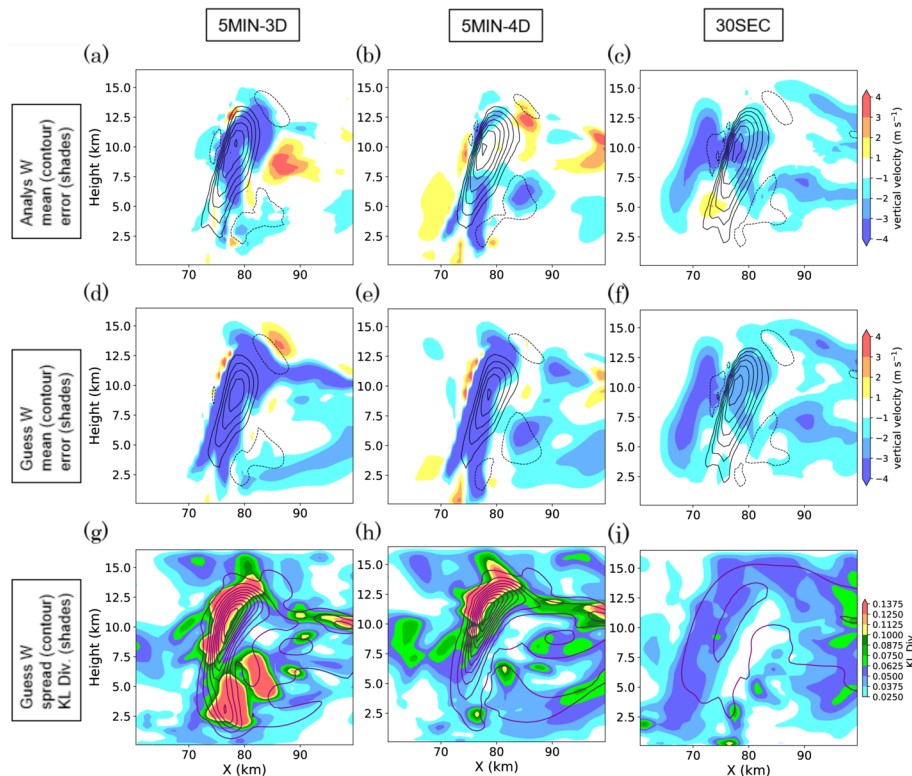

**Figure 11.** Same with figure 9 but for the experiments with background stability perturbation.

ensemble mean precipitation forecasts have more realistic amplitude of errors than the case with the true profile. The wind and stability perturbations produce characteristic spatial patterns in the forecast error. Meanwhile, the difference among the three data assimilation methods does not cause significant differences in the forecast precipitation, despite the difference in the analysis errors in vertical velocity fields.

## 5 Summary and discussions

In this study, the idealized experiments were conducted to examine the impact of assimilating radar reflectivity every 30 seconds on the non-Gaussianity of first-guess error distribution and the analysis and forecast accuracy. As we focused on convective scale errors, we designed the experiment to exclude other factors such as errors in the forecast model and the observation operator, uncertainties in background vertical profiles of atmospheric variables, and modifications on the background error covariance in the EnKF such as covariance inflation.

We performed perfect-model OSSEs of an idealized supercell development from a warm bubble. Synthetic radar reflectivity observation data was created from the time series of the nature run every 30 seconds. We compared the analyses produced by the data assimilation cycles with three different manners, namely, 5-minute 3D-LETKF (5MIN-3D), 5-minute 4D-LETKF

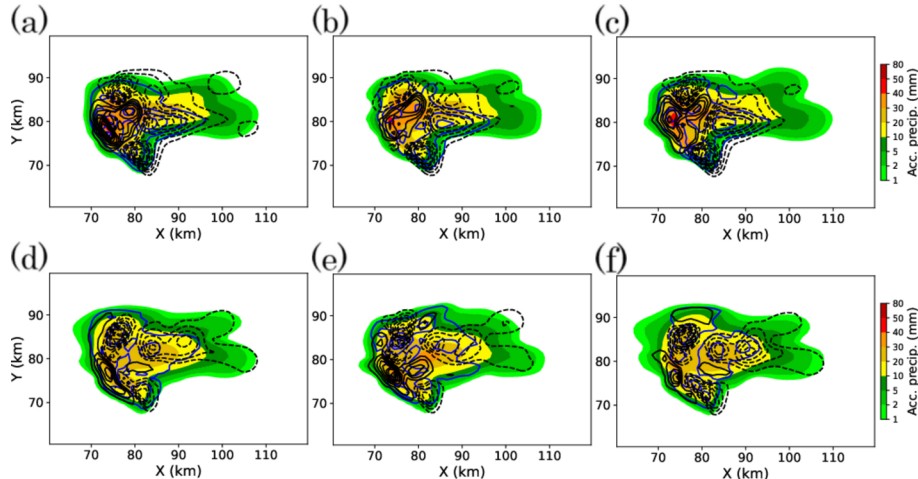

**Figure 12.** Ensemble mean 30-minute accumulated surface precipitation of forecasts from 00:50:00, of the experiments with (a-c) background wind and (d-f) background stability perturbation. They are of (a,d) 5MIN-3D, (b,e) 5MIN-4D, and (c,f) 30SEC cases respectively. Black contours indicate the difference from the nature run with a contour interval of 2 mm. Blue contours indicate the ensemble spread in surface precipitation with a contour interval of 2 mm.

(5MIN-4D), and 30-second 3D-LETKF (30SEC). We found a significant reduction of the non-Gaussianity for vertical velocity in 30SEC compared to other cases, along with the reduction of the ensemble spread. We also found the improvement of the analysis accuracy for vertical velocity in 30SEC. The impact was larger when compared with 5MIN-3D. Smaller but still significant differences were found when compared to 5MIN-4D.

315    The significant difference between the cases was found mostly in the upper part of the main convective cell, where the vertical velocity has the largest value. In contrast, the dynamical variables in the lower levels that mainly control the evolution of the supercell system did not change significantly. The ensemble mean reflectivity and surface accumulated precipitation of 30-minute forecast from the analysis ensemble did not show significant differences between the cases.

We further examined the first guess ensemble where significant non-Gaussianity is found. We compared the joint distribution
320    between graupel mixing ratio and vertical velocity among the three cases and found a significant difference in the relationship between the two variables. In 5MIN-3D, not only the non-Gaussianity of the background error of a single variable but also the nonlinearity in the approximate relationship between the two variables was found, whereas in 30SEC they were significantly reduced. We suggested the possible impact of this nonlinearity to the difference of analysis error in vertical velocity among the three cases, besides the difference of first guess ensemble spread.

325    We concluded that assimilating radar reflectivity every 30 seconds indeed has a significant impact on the analysis accuracy for unobserved variables, and that it could be caused by the joint non-Gaussian background error probability density of multiple variables. However, at the same time, we also found that the importance of the analysis accuracy improvement is not significant for short-period precipitation forecasting, as it does not essentially change the evolution of the convective system itself. The

30-minute forecast of 5MIN-3D already showed a highly accurate precipitation forecast, suggesting that radar reflectivity of every 5 minutes is sufficient in the idealized setting of this study, where errors from the model and observation operator, and the uncertainty in larger-scale atmospheric fields are all ignored. Therefore, where the impact of non-Gaussianity is considered separately, it is likely to be less significant than other factors.

We further performed a similar comparison under more realistic situations where we have uncertainty at a larger scale. To mimic realistic settings, 10 different background profiles were used to create a 100-member initial ensemble. In the two experiments, background wind and stability profiles were perturbed, respectively. These experiments showed a more significant difference between assimilation frequencies of 5 minutes and 30 seconds in first guess vertical velocity. The biases in background profiles caused larger deviations among first guess members and thus large non-Gaussianity during 5-minute integration. On the other hand, the 30SEC case has significantly reduced non-Gaussianity. It produced significantly smaller analysis errors than 5MIN-3D. However, in these experiments, we also found more complex features in first guess and analysis errors which we did not find in the previous experiment. First, the analysis error of 5MIN-4D has patterns that are significantly different from those of 5MIN-3D. This might be caused by 4D-LETKF method which attempts to optimize the time series within the window instead of the instantaneous analysis value. Second, we found some patterns in first guess and analysis error in 30SEC which were not seen in the others. This was supposed to be caused by frequent data assimilation and possibly driven by the accumulation of unbalanced analysis increment. The issue of imbalance caused by frequent data assimilation with ensemble Kalman filter was discussed in previous studies such as He et al. (2020). A recent work by Huo et al. (2025) discusses the application of incremental analysis updates to tackle this issue in the context of every 30 seconds assimilation of PAWR observation.

The findings in these additional experiments may provide more insights into the interpretation of previous studies. For example, in the experiment using the Osaka PAWR data (Ruiz et al., 2021), they found a significant difference in first guess mean vertical velocity not only in magnitude but also in structure (their figures 2a and 2d). The situation may be explained by the case shown in Figure 10, indicating that the large uncertainty in background thermal profile might cause a large difference in analysis vertical velocity field.

We changed nothing in the LETKF when we performed the experiments with perturbed background profiles, for comparison with the previous studies. However, assimilating the observation only with a small localization scale was inefficient in constraining larger-scale fields and did not improve the forecast accuracy, which is supposed to be mostly driven by larger-scale atmospheric states. We can consider more advanced approaches to deal with this problem. Some studies proposed methods of multi-scale data assimilation (Zhang et al., 2009; Miyoshi and Kondo, 2013; Fabry, 2022). Also, finding the optimal parameter of covariance inflation, RTPS, or RTPP would have a significant impact in this case, as it compensates the underestimation of model error. The application of those methods to frequent radar data assimilation problems will be examined in future studies.

Data assimilation methods without the assumption of Gaussianity, such as particle filters (van Leeuwen et al., 2019), are another potential approach to deal with the problem of non-Gaussianity. Particle filters can avoid the error caused by linear superposition when the errors in the state variables have nonlinear relationships, as seen in Figure 6a. Therefore, this approach has the potential to improve the analysis accuracy in such situations.

*Code and data availability.* The SCALE-LETKF code is available on Zenodo: https://doi.org/10.5281/zenodo.17156801 (Amemiya et al.
2025). All the code and data to reproduce the figures in this study are available on request from the corresponding author.

*Author contributions.* AA was responsible for conceptualization, formal analysis, methodology, investigation, visualization, and writing.
TM was responsible for the project administration, investigation, and review and editing of the manuscript.

*Competing interests.* Some authors are members of the editorial board of journal NPG.

*Acknowledgements.* We thank the editor and the anonymous referees for their constructive comments. This study is supported by JSPS KAK-
ENHI (Grant No. JP21K13996 and JP24H00021), JST SATREPS (Grant number: JPMJSA2109), JST CREST Grant Number JPMJCR24Q3,
Japan Aerospace Exploration Agency (JAXA), RIKEN Transformative Research Innovation Platform (TRIP) Project "Prediction Science",
and the Center of Excellence (COE) research grant in computational science from Hyogo Prefecture and Kobe City through the Founda-
tion for Computational Science (FOCUS). This study used the computational resource of the HPCI general access project hp230094 and
hp240061. Some of the figures are made using the MetPy library (https://doi.org/10.5065/D6WW7G29).

365
370

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
