# Peer review of "Impact of reduced non-Gaussianity on analysis and forecast accuracy by assimilating every-30-second radar observation with ensemble Kalman filter: idealized experiments of deep convection"

_EGUsphere, 2025_

## Author Comment (AC2)

[Figure]

**Figure 4.** Vertical cross sections at Y = 80 km of (a-c) analysis ensemble mean and deviation from the nature run, (d-f) first guess ensemble mean and deviation from the nature run, and (g-i) first guess ensemble spread and KL divergence, of vertical velocity at 00:50:00, for (a,d,g) 5MIN-3D, (b,e,h) 5MIN-4D, and (c,f,i) 30SEC cases respectively. Black contours in (a-f) indicate ensemble mean of vertical velocity with contour interval of 5 m s$^{-1}$. Red contours in (g-i) indicate first guess ensemble spread of vertical velocity with contour interval of 0.5 m s$^{-1}$.

[Figure]

**Figure 9.** Same with figure 4 but for the experiments with background wind perturbation. Purple contours in (g-i) indicate first guess ensemble spread of vertical velocity with contour interval of 1 m s$^{-1}$.

[Figure]

**Figure 10.** Same with figure 9 but for the experiments with background stability perturbation.

---

## Author Response (AR1)

[Response to RC1 by Dr. Wei Han, 10 Jul 2025]

(the original referee comments are denoted in blue and italic texts)
>This paper examines the effect of assimilating high-frequency radar observations on analysis and forecast accuracy in convection-permitting numerical weather prediction. The authors conduct idealized experiments using the local ensemble transform Kalman filter (LETKF) and find that, compared to a 5-minute assimilation interval, assimilating radar reflectivity every 30 seconds significantly reduces non-Gaussianity in the background error distribution and improves analysis accuracy, especially for vertical velocity. However, it does not significantly improve precipitation forecasts. Additionally, the study offers several insights into the initial perturbation scheme. The paper is well-organized but could be improved by addressing the following issues:

We thank very much Dr. Wei Han for the referee comments. We answer each comment in the following.

>Possible typographical and grammatical errors:
>Line 112: 'assimilation or Doppler velocity is also not considered' should be changed to 'assimilation of Doppler velocity is also not considered.'
>The caption for Table 1 should appear above the table, not below it.
>Line 140: 'figures 1a-1c' should be corrected to 'figures 3a-3c.'
>The description for the sub-figures in the third row (e.g., Figures 4, 9, and 10) is not clearly presented.
>Line 221: '5min-3D' should be replaced with '5MIN-3D' for consistency.
>In the caption for Figure 9, either the color of the contours should be changed to purple, or the word 'Purple' should be changed to 'Red' to match the figure.

Thank you for pointing them out.

Regarding line 140 'figures 1a-1c', we meant 'figures 2a and 2c', the horizontal and vertical cross sections of the nature run at the time. We have revised that part of the sentence.

Regarding the inconsistency between the caption and image of Figure 9, we found that we applied the contour color and interval setting which was used for Fig.4, although we intended to change them as written in the caption. We have revised Fig.4, Fig.9 and Fig.10 to fix the color shading and show the color map title (KL div.).

For the rest, we have corrected them as suggested.

>General recommendations:
>Given the design of these idealized experiments, 100 ensemble members are sufficient to

*reduce sampling error to a small degree. However, it would be beneficial to include a discussion on the impact of ensemble size on sampling error, or at least cite relevant previous studies in this area.*

We considered that using 100 ensemble members is sufficient given the small localization scale. We set 4 km and 2 km for horizontal and vertical localization length scale respectively, while we had 1 km and 200 m horizontal and vertical grid spacings, which correspond to at least 4 km and 800 m resolution, respectively, (according to https://glossary.ametsoc.org/wiki/Model_resolution). Therefore the rough estimate of the effective degree of freedom of a localized ensemble background field is 2x2x5=20 for one variable. Therefore, we considered the ensemble size of 100 is larger or at least comparable to the effective degree of freedom of localized background error.

To include this discussion, we have revised the sentence in lines 141-143 in the revised manuscript.

*>It is reasonable to disable temporal localization for the 5MIN-4D case to ensure a fair comparison with the 30SEC case. However, it should be noted in the discussion that assimilating observations every 30 seconds is not currently practical in real-world operational systems. The aim of this study is to explore the underlying relationship between assimilation frequency and non-Gaussianity in an idealized setting, rather than to propose a practical assimilation strategy.*

We mentioned it lines 51-53 in the introduction. We have also added the following sentence after line 152 of Section 2.3 in the revised manuscript.

"Although this choice might not be practical, we prioritize exploring the underlying relationship between assimilation frequency and non-Gaussianity in an idealized setting."

*>The impact of assimilation frequency in idealized experiments has been previously discussed by [1]. If possible, please provide a theoretical conclusion regarding the effect of assimilation frequency under the EnKF framework for non-Gaussian problems.*

Thank you for introducing us the related previous study. This study focuses on timescales between 30 seconds and 5 minutes, which is much smaller than 1 to 6 hours discussed in the previous study. The process which adjusts imbalance is not gravity waves but acoustic waves and possibly moist convection. Therefore it is not straightforward to compare the findings in this study with the previous study, but it can be said that the more frequent (30-second) assimilation in this study was shown to be advantageous in improving analysis and subsequent forecast (next first guess) accuracy, as shown in Fig.4. This is thought to be due to smaller analysis increment at each step, causing less nonlinear error growth. I have added the article

[1] and another relevant article to the reference list and added the corresponding discussion in lines 326-329 in Section 5 in the revised manuscript.

>*This study shows that increasing assimilation frequency improves the analysis state but does not significantly improve forecast performance. Is this a coincidental result, or have similar findings been reported in other studies?*

We consider the same result has not been reported in other studies, as not many existing studies addressed this topic focusing at a short time scale as 30 seconds assimilation cycle and 30 minutes forecast. However, we consider we can interpret this result on the analogy of a similar issue at longer time scales, which we have a consensus. In general, the impact of improved initial condition by data assimilation is dominant in earlier forecast time and overwhelmed by the impact of boundary conditions in later forecast time (mentioned in [2] for example). Then we expect that the accuracy of 30 minutes forecast is more controlled by a larger-scale atmospheric variable field, which is not significantly constrained by data assimilation with a small localization scale.

Reference:

[1] He, Huan, et al. "Impacts of assimilation frequency on ensemble Kalman filter data assimilation and imbalances." Journal of Advances in Modeling Earth Systems 12.10 (2020): e2020MS002187.
[2] Clark, Peter, et al. "Convection-permitting models: A step-change in rainfall forecasting." Meteorological Applications 23.2 (2016): 165-181.

[Response to RC2 by Dr. Zheqi Shen, 26 Jul 2025]

*> The study investigates the impact of high-frequency radar data assimilation on analysis and forecasting accuracy, which is a topic of significant scientific and practical importance. The experimental design is rational and rigorous. By conducting idealized experiments, the authors successfully eliminate complex interfering factors present in real-world applications, thereby enhancing the credibility of their findings. The results demonstrate that assimilating radar data every 30 seconds can significantly reduce non-Gaussianity and improve the analysis accuracy of vertical velocity. These findings provide valuable insights for future research on radar data assimilation. Overall, the paper addresses a meaningful topic, features a well-designed experiment, and presents reliable results. It is recommended for publication. I suggest a minor revision.*

*> The other reviewer, Dr. Wei Han, has already pointed out some details regarding the figures and several important general opinions, with which I fully agree. Here, I would like to add some of my personal concerns.*

We thank Dr. Zheqi Shen very much for the referee comments. We answer each comment in the following.

*> Presentation of Assimilation Results: The discussion of the assimilation results, such as in Figures 3, 4, 9, and 10, only shows the final assimilation at 00:50:00. While a single assimilation can demonstrate the improvement effects of different schemes, completely ignoring the entire assimilation process seems inappropriate. I suggest using time series of some metric (such as RMSE or spread) or showing errors and spread at several different moments to illustrate how the assimilation gradually takes effect and reaches stability.*

We agree on the importance of confirming the evolution of metrics over the period of data assimilation cycle. However, in this experiment, the deep convection rapidly develops and makes the concept of stability difficult to apply, because the number of assimilated observations, the value of maximum reflectivity, and the area of high reflectivity all evolve with time. The figure in the file "supplement_figureA_AC3.pdf" (attached to AC3) shows the evolution of domain-averaged RMSE and spread in reflectivity, number of assimilated observations, and maximum analysis vertical velocity in the case of 5MIN-3D. The assimilated observation has a peak at 20 minutes, as 'no precipitation' observation signals are assimilated to remove artificial convection at random locations triggered by the initial random perturbation in some members. As the area of high reflectivity unfolds, the observation number increases again after about 50 minutes, and RMSE and spread keep increasing. The maximum value of analysis mean vertical velocity reaches around the peak value of 40 m/s at

50 minutes. Therefore, we focus on the analysis currently, considering that the data assimilation has run enough to ignore the effect of initial adjustment, although the metrics do not show convergence.

Regarding this issue, we have added the sentences in lines 167-170 in Section 2.5 of the revised manuscript.

> *Terminology in EnKF Context: I feel that the term "first-guess" is more commonly used in variational assimilation. In the context of EnKF assimilation, "prior" might be more suitable. This is just a personal suggestion.*

We understand that using "prior" is more common in some groups in the EnKF community, while there are papers which chose to use "first-guess". As this study addresses the issue of non-Gaussianity, which was discussed in earlier studies such as Ruiz et al. 2021, which used "first-guess", we decide to use "first-guess" in this paper for consistency.

> *Temporal Localization (Line 129): I am not quite familiar with the term "temporal localization." Does it equate to the description of using different weights for observations at different times? The 5MIN-4D scheme not only uses ten times the amount of data compared to 5MIN-3D but also assigns all data from different times to the 5th minute without increasing the standard deviation of observational errors. I think the current description is not detailed enough and should be improved for the 5MIN-4D scheme.*

The idea of temporal localization is to impose a weighting factor which is a function of the time difference within the assimilation window, effectively changing the relative observation error. As you pointed out, in the 5MIN-4D case, when temporal localization is not used, all the observations are assimilated with the same prescribed observation error standard deviation. We have revised lines 142-144 in the revised manuscript, adding clearer and more detailed description, considering this and the other referee's comment.

> *Introduction to LETKF: Although LETKF is a very well-known method, I believe it is necessary to briefly introduce LETKF in the section on the assimilation system, especially how the Gaussian assumption is embedded in its algorithm.*

We agree with your suggestion. We have added the introduction to LETKF after the first paragraph of Section 2.3 in the revised manuscript.

> *Inflation Setting (Line 75): I agree with the no-inflation setting, as inflation under different assimilation frequencies can significantly affect the spread. I think a discussion on the possible*

*impact of inflation on the conclusions in practical scenarios could be added to the conclusion section.*

We have added lines 339-341 in Section 5 in the revised manuscript to cover this discussion.

*> Figure Presentation: Figures 4, 9, and 10 contain very rich information, requiring repeated reading between the text and the figures. I suggest adding the names of the experiments to the titles of subplots (a), (b), and (c) to facilitate reading. Moreover, the contour lines in Figures 9 and 10 are too thin and light in color, making them hard to see. They need to be improved. The shading information also needs to be displayed in softer colors or with increased transparency.*

Thank you for the important suggestion. We have revised Figures 4, 9, and 10, using lighter colormap for the shading.

*> Clarification on Perturbations (Line 210): The description of perturbations is not clear. It appears that there are 10 background wind profiles (or background thermal profiles) perturbed, resulting in 100 members. This seems to be combined with the initial perturbation scheme in Section 2.4. What is unclear to me is whether the 10 perturbations are superimposed on the 100 perturbations from Section 2.4, or whether the 10 perturbations from Section 2.4 are combined with these 10. It is necessary to clarify how the 100 members are generated, rather than just stating the conclusion: "Both of those 10 sets include one true profile, indicating only 10 members have the correct background wind or stability profile. The other 90 members are biased and are expected to have significant errors in the evolution at a convective scale."*

The 100 perturbations described in Section 2.4 are imposed in the same way. We have added the following sentence in the first paragraph of Section 4.1.

"The random perturbation described in Section 2.4 (Table 1) is imposed on each ensemble member in the same way as before."

---

## Referee Report (RR1)

egusphere-2025-2543

Impact of reduced non-Gaussianity on analysis and forecast accuracy by assimilating every-30-second radar observation with ensemble Kalman filter: idealized experiments of deep convection

Arata Amemiya and Takemasa Miyoshi

General comments:

This study conducted observation system simulation experiments (OSSEs) idealized for high-frequency radar reflectivity data assimilation (DA) with local ensemble transform Kalman filter (LETKF) and compared 3 experiments: (i) 3D-LETKF every 5 min, (ii) 4D-LETKF every 5 min with observations every 30 sec, and (iii) 3D-LETKF every 30 sec. As a result, (iii) had the smallest non-Gaussianity of first guess ensemble and the best analysis accuracy. Since it is interesting to clarify the advantages of high-frequency DA with the idealized OSSEs, this study is valuable to be published. However, the causal relationship between the non-Gaussianity of first guess ensemble and the analysis accuracy is not clear even in the OSSEs conducted in this study. Therefore, it should be clarified that this study does not investigate pure impact of non-Gaussian distribution but imply the advantages of high-frequency DA partially in the viewpoint of non-Gaussianity. To prevent misunderstanding about it, I recommend major revision. The order of the following comments is not related to importance.

Specific comments:

1. L4-5, L50-51, L75-77, and L286-287: The OSSEs conducted in this study do not completely exclude the impact other than non-Gaussianity because the DA method, the number of assimilated observations, the ensemble spread, etc. are different between the 3 experiments compared in this study. This study does not clarify the pure impact of non-Gaussian distribution but imply the advantages of high-frequency DA partially in the viewpoint of non-Gaussianity. The sentences should be revised not to be misunderstood.

2. L20-23: This description is not necessarily correct in 4D-EnKF. Since the experiments with 4D-EnKF are conducted in this study as well as 3D-EnKF, the development of 4D-EnKF and its advantage also should be explained.

3. L53-54: Why is this study useful for the non-Gaussian DA? Could you cite any previous studies?

4. L73-77 and L288-290: This experimental design does not completely exclude modification to the ensemble perturbations because spatial localization is applied to reduce the effect of sampling error. This limitation should be stated.

5. Figure 1: To add axis of height (km) is helpful to understand the height of convections shown in other figures.

6. L90-91: The forcing by the warm bubble should be stated more concretely and quantitatively here.

7. L137-138: Does it mean that 5-dBZ reflectivity is assimilated even where the first guess < 5 dBZ? In this case, the precipitation becomes stronger in the analysis. Is it no problem?

8. L143-146: This description is redundant and difficult to be understood. Does it mean that reflectivity is assimilated only where at least one ensemble member > 10 dBZ in the first guess?

9. L160-161: Why was the potential temperature perturbation over the entire domain in addition to the perturbation in the warm bubble?

10. L161-162: Were the Gaussian perturbations added in the warm bubble and the whole domain at the first assimilation cycle? If so, why is the first guess ensemble expected to be non-Gaussian? How to add the perturbations should be explained more clearly.

11. L183: What determines this kernel bandwidth? Could you cite any previous studies?

12. L191-192: The ensemble spread in the 5MIN-4D case should also be stated. In addition, it is better to show the time series of the ensemble spread until 00:50:00 in all cases to confirm that the filter divergence has not occurred.

13. Figure 5: To make the discussion in Section 3.2 deeper, the ensemble spread should also be shown in addition to ensemble mean and the difference from the nature run. I think main difference between the 3 experiments is the ensemble spread.

14. L213-214, L276-277, L279-280, L283-284, and L309-314: If the ensemble spread or the KLD is largely different between the 3 experiments, these descriptions are not precise for ensemble forecasts. The difference should be shown also for the ensemble forecasts.

15. Figure 6: The time of the 100 samples should be shown as well as the position of the grid point.

16. L221-223 and L304-306: The nonlinear cross-variable relationship between graupel mixing ration and vertical velocity looks caused by difference of non-Gaussianity of these variables (vertical velocity is closer to Gaussian). If this interpretation is correct, it should be stated.

17. L227-228: Could you show the mathematical definition of "the mutual information between the ensemble members of graupel and vertical velocity at every grid point, after removing the linear dependency"?

18. Figure 7: The caption should include "in the 5MIN-3D case".

19. L230-231 and L306: The nonlinear relationship between the two variables may be one cause of the error of the analysis. However, it may simply be caused by smaller impact of assimilation in the part of smaller ensemble spread. Or, the smaller number of observations assimilated with low-frequency may also make the large error of the analysis. It is better to show various possible causes.

20. L237: "pertrbed" -> "perturbed"

21. L253-255, L266-269, L323-324, and L325-326: These disadvantage of 5MIN-4D and 30SEC are general and should be found also in the experiments in Section 3. Why are they found only in the additional experiments in Section 4?

---

## Referee Report (RR2)

egusphere-2025-2543

Impact of reduced non-Gaussianity on analysis and forecast accuracy by assimilating every-30-second radar observation with ensemble Kalman filter: idealized experiments of deep convection

Arata Amemiya and Takemasa Miyoshi

Thank you for revising the manuscript properly based on my comments. Although many of my concerns are addressed, I suggest the authors to address the following minor comments additionally. The order of the following comments is not related to importance.

Comments:

1. L23-26: Could you add any references for 4D-EnKF?

2. L61: The unnecessary "re" should be removed.

3. L77-79 and L314-317: The spatial localization modifies the form of probability distribution function of the ensemble perturbations in EnKF analysis. It should be stated as one of the limitations of this study.

4. L128: "$\mathbf{y}^b$" -> "$\bar{\mathbf{y}}^b$"

5. L146-148: The descriptions of the threshold and the upper limit are not necessary because they are obvious or redundant. (I don't think the number of observations assimilated at a grid point exceeds 100.)

6. L162-163: This study states "we assume only convective-scale uncertainty (L73)". Nevertheless, why are the perturbations necessary in the area outside the convective cell?

7. L164-165: It should be stated that the time when high reflectivity is first observed is 10 min after the initial Gaussian perturbations are added.

8. Table 1: The height of the center location also should be written here.

9. L191-194: To describe the definition of p(x,y) may help readers understand.

10. Figure 3:
    i.    The time after 00:50:00 is confusing and not necessary to be plotted because it is not the data assimilation period.
    ii.   The assimilation window is not 5 min in 30SEC. Nevertheless, why the lines of 30SEC are plotted every 5 min? They should be plotted every 30 sec. The consistency to the explanation in L211-214 also should be confirmed.

11. L201: "The first The number of assimilated observation peaks …" -> "The first peak of the number of assimilated observations is …"

12. L212-213: "respectively6.6 dBZ in the 5MIN-3D case and 3.2 dBZ in the 30SEC case" -> "respectively"

13. L235-237 and L304-307: Why was the impact on ensemble forecasts limited although analysis ensemble spreads were significantly different? To answer this question is important for this study.

14. Figure 7: Could you state the position (X = 78 km, Y = 80 km, Z = 12.1 km) also in the caption?

15. L245-246: Could you explain your interpretation that nonlinearity appears as non-Gaussianity also in the text?

16. L277-280, L290-292, and L349-354: Why the biased background profile causes these side effects should be explained briefly in the text.

17. L361: "Figure 10" -> "Figure 11"
18. L372: "Figure 6a" -> "Figure 7a"

---

## Author Response (AR2)

[Response to referee report by Anonymous Referee #3, 21 Sep 2025]

(the original referee comments are denoted in blue and italic texts)

*This study conducted observation system simulation experiments (OSSEs) idealized for high-frequency radar reflectivity data assimilation (DA) with local ensemble transform Kalman filter (LETKF) and compared 3 experiments: (i) 3D-LETKF every 5 min, (ii) 4D-LETKF every 5 min with observations every 30 sec, and (iii) 3D-LETKF every 30 sec. As a result, (iii) had the smallest non-Gaussianity of first guess ensemble and the best analysis accuracy. Since it is interesting to clarify the advantages of high-frequency DA with the idealized OSSEs, this study is valuable to be published. However, the causal relationship between the non-Gaussianity of first guess ensemble and the analysis accuracy is not clear even in the OSSEs conducted in this study. Therefore, it should be clarified that this study does not investigate pure impact of non-Gaussian distribution but imply the advantages of high-frequency DA partially in the viewpoint of non-Gaussianity. To prevent misunderstanding about it, I recommend major revision. The order of the following comments is not related to importance.*

We appreciate the referee for the valuable comments. We have revised the manuscript accordingly. Below are the responses to each comment.

Specific comments:

*1. L4-5, L50-51, L75-77, and L286-287: The OSSEs conducted in this study do not completely exclude the impact other than non-Gaussianity because the DA method, the number of assimilated observations, the ensemble spread, etc. are different between the 3 experiments compared in this study. This study does not clarify the pure impact of non-Gaussian distribution but imply the advantages of high-frequency DA partially in the viewpoint of non-Gaussianity. The sentences should be revised not to be misunderstood.*

We appreciate the important comment. We agree that the experimental design did not exclude the impact of the difference in ensemble spread among the three cases, which is significant in some parts of the domain. Also we agree that the number of assimilated observations is different between 4D-5MIN and 30SEC because of the threshold of gross error and the number of first guess ensemble members for the assimilation. We have rewritten the sentences to make the scope of this study clearer (L4-5, L53-54, L77-79, and L314-317 in the revised manuscript).

*2. L20-23: This description is not necessarily correct in 4D-EnKF. Since the experiments with 4D-EnKF are conducted in this study as well as 3D-EnKF, the development of 4D-EnKF and its advantage also should be explained.*

I have added the description of 4D-EnKF (L23-26 in the revised manuscript).

*3. L53-54: Why is this study useful for the non-Gaussian DA? Could you cite any previous studies?*

I have rewritten the sentence to make the implication clearer (L55-56 in the revised manuscript).

*4. L73-77 and L288-290: This experimental design does not completely exclude modification to the ensemble perturbations because spatial localization is applied to reduce the effect of sampling error. This limitation should be stated.*

In this sentence we meant to focus on the ensemble perturbation as an approximation of the background probability distribution. We think it is not the ensemble perturbation but the spatial structure of background error correlation that is modified by spatial localization.

*5. Figure 1: To add axis of height (km) is helpful to understand the height of convections shown in other figures.*

We have revised Figure 1 adding the secondary vertical axis showing height.

*6. L90-91: The forcing by the warm bubble should be stated more concretely and quantitatively here.*

It was stated in the sentences which follow L90-91. We have rephrased the description to make it clear (L92-95 in the revised manuscript).

*7. L137-138: Does it mean that 5-dBZ reflectivity is assimilated even where the first guess < 5 dBZ? In this case, the precipitation becomes stronger in the analysis. Is it no problem?*

It does not occur as the adjustment to 5 dBZ is also applied to the first guess. We have added the explanation about it (L139-141 in the revised manuscript).

*8. L143-146: This description is redundant and difficult to be understood. Does it mean that reflectivity is assimilated only where at least one ensemble member > 10 dBZ in the first guess?*

We have rephrased the description and moved it to the earlier part just after the description of the treatment of reflectivity (L141-142 in the revised manuscript).

*9. L160-161: Why was the potential temperature perturbation over the entire domain in addition to the perturbation in the warm bubble?*

It was applied to add nonzero spread of the dynamical variables in the area outside the convective cell. We have added the explanation (L162-164 in the revised manuscript).

*10. L161-162: Were the Gaussian perturbations added in the warm bubble and the whole domain at the first assimilation cycle? If so, why is the first guess ensemble expected to be non-Gaussian? How to add the perturbations should be explained more clearly.*

In the previous version of the manuscript, the expression "the first guess ensemble *at the first data assimilation cycle* is expected to have ..." was not clear and confusing.

In fact, the first time step when the radar reflectivity is assimilated is 00:10:00, as it takes about 10 minutes from the initial time for the convective cell to develop enough to be observed. The expected non-Gaussianity comes from the development of perturbation for that initial 10 minutes interval.

We have revised the description (L164-165 in the revised manuscript). Also, as the response to the comment 12 below, we have added Fig.3 to show the time series of the number of assimilated observations.

*11. L183: What determines this kernel bandwidth? Could you cite any previous studies?*

The equation (9) is derived in the Silverman (1986) textbook and commonly used for kernel density estimation. I have added the reference.

*12. L191-192: The ensemble spread in the 5MIN-4D case should also be stated. In addition, it is better to show the time series of the ensemble spread until 00:50:00 in all cases to confirm that the filter divergence has not occurred.*

We have added the maximum value of the ensemble spread in the 5MIN-4D case in the text (L211-212 in the revised manuscript). We have also added Fig.3, which shows the time series of the total number of assimilated observations in 5-minutes window and ensemble spread in reflectivity averaged over the grid points where the true reflectivity value is over 10dBZ.

*13. Figure 5: To make the discussion in Section 3.2 deeper, the ensemble spread should also be shown in addition to ensemble mean and the difference from the nature run. I think main difference between the 3 experiments is the ensemble spread.*

We have added the contours of ensemble spread in reflectivity and surface accumulated precipitation in Fig. 5 and 11 (Fig.6 and 12 in the revised manuscript).

*14. L213-214, L276-277, L279-280, L283-284, and L309-314: If the ensemble spread or the KLD is largely different between the 3 experiments, these descriptions are not precise for ensemble forecasts. The difference should be shown also for the ensemble forecasts.*

The ensemble spreads of the three experiments shown in Fig.5 and 11 (Fig.6 and 12 in the revised manuscript) are mostly not different from each other, though they are different in analysis reflectivity and vertical velocity. It further supports the conclusion in those sections that the different data assimilation frequency has a limited impact on precipitation forecast. We have added the additional sentences (L235-237 and L304-306 in the revised manuscript).

*15. Figure 6: The time of the 100 samples should be shown as well as the position of the grid point.*

We have added the time to the caption of Fig. 6 (Fig.7 in the revised manuscript).

*16. L221-223 and L304-306: The nonlinear cross-variable relationship between graupel mixing ration and vertical velocity looks caused by difference of non-Gaussianity of these variables (vertical velocity is closer to Gaussian). If this interpretation is correct, it should be stated.*

Our interpretation is rather that nonlinearity appears as non-Gaussianity. When there is a nonlinear relationship between two variables caused by the nature of the system dynamics, they have different degrees of non-Gaussianity in the probability distribution, even though one of them may be nearly Gaussian.

*17. L227-228: Could you show the mathematical definition of "the mutual information between the ensemble members of graupel and vertical velocity at every grid point, after removing the linear dependency"?*

We have added the description and mathematical formula to calculate it in Section 2.5.

*18. Figure 7: The caption should include "in the 5MIN-3D case".*

We have revised the caption as suggested.

*19. L230-231 and L306: The nonlinear relationship between the two variables may be one cause of the error of the analysis. However, it may simply be caused by smaller impact of assimilation in the part of smaller ensemble spread. Or, the smaller number of observations assimilated with low-frequency may also make the large error of the analysis. It is better to show various possible causes.*

We agree that the impact of the difference of first guess ensemble spread in vertical velocity may be significant and the impact of nonlinearity can't be easily evaluated separately from it. We have revised the sentences to mention both possibilities (L255-256 and L333-334 in the revised manuscript).

*20. L237: "pertrbed" -> "perturbed"*

We have corrected the typo.

*21. L253-255, L266-269, L323-324, and L325-326: These disadvantage of 5MIN-4D and 30SEC are general and should be found also in the experiments in Section 3. Why are they*

*found only in the additional experiments in Section 4?*

We have a speculation that the side effect appears in those experiments because of the large bias in the first guess caused by a biased background profile, which is the main difference from the experiment in Section 3.